# Nucleosome wrapping energy in CpG islands and the role of epigenetic base modifications

Rasa Giniūnaitė[1,2], Rahul Sharma[3], John H Maddocks[3], Skirmantas Kriaucionis[4], Daiva Petkevičiūtė-Gerlach[1]*

[1]Department of Applied Mathematics, Kaunas University of Technology, Studentų, Kaunas, Lithuania; [2]Institute of Applied Mathematics, Vilnius University, Naugarduko, Vilnius, Lithuania; [3]Institut de Mathématiques, École Polytechnique Fédérale de Lausanne, EPFL SB MATH LCVMM, Lausanne, Switzerland; [4]Ludwig Institute for Cancer Research, University of Oxford, Nuffield Department of Medicine, Old Road Campus Research Building, Roosevelt Drive, Oxford, United Kingdom

## eLife Assessment

This **valuable** simulation study proposes a new coarse-grained model to explain the effects of CpG methylation on nucleosome wrapping energy. The model accurately reproduces the all-atom molecular dynamics simulation data, and the evidence to support the claims in the paper is **solid**. This work will be of interest to researchers working on gene regulation, mechanisms of DNA methylation and effects of DNA methylation on nucleosome positioning.

*For correspondence:
daiva.petkeviciute@ktu.lt

Competing interest: The authors declare that no competing interests exist.

**Abstract** The majority of vertebrate promoters have a distinct DNA composition, known as a CpG island. Cytosine methylation in promoter CpG islands is associated with a substantial reduction of transcription initiation. We hypothesise that both atypical sequence composition and epigenetic base modifications may affect the mechanical properties of DNA in CpG islands, influencing the ability of proteins to bind and initiate transcription. In this work, we model two scalar measures of the sequence-dependent propensity of DNA to wrap into nucleosomes: the energy of DNA required to assume a particular nucleosomal configuration and a measure related to the probability of linear DNA spontaneously reaching the nucleosomal configuration. We find that CpG density and modification state can alter DNA mechanics by creating states more or less compatible with nucleosome formation.

## Introduction

CpG islands (CGIs) are regions in vertebrate genomes with a higher frequency of CpG dinucleotide steps (*Bird et al., 1985*; *Gardiner-Garden and Frommer, 1987*) than surrounding DNA. This is a reflection of the general depletion of CpGs outside CGIs, where CpGs are observed at around one fifth of the randomly expected frequency (*Lander et al., 2001*). Most vertebrate, including human, genes often have associated CGIs (*Cooper et al., 1983*; *Larsen et al., 1992*) typically coinciding with sites of transcription initiation and likely contributing to the regulation of gene activity (*Deaton and Bird, 2011*). One way CGIs function is by attracting chromatin proteins with the CxxC domain, which recognise epigenetically unmodified CpGs and are instrumental for the establishment of characteristic chromatin modification profiles at CGIs (*Long et al., 2013a*).

**Table 1.** Total number of considered sequences from different regions of the human genome.
Sequence listings are available at https://github.com/rginiunaite/CGI-NMI-sequences.

|  | NMI | Not NMI | Total |
|---|---|---|---|
| CGI | 20257 | 7413 | 27670 |
| Not CGI | 16855 | 42349 | 59204 |
| Total | 37112 | 49762 | 86874 |

The general consensus is that the majority of CGIs are epigenetically unmodified, whereas in the regions outside CGIs most cytosines in the CpG dinucleotides are methylated (*Ioshikhes and Zhang, 2000*; *Hannenhalli and Levy, 2001*; *Bock et al., 2007*; *Han and Zhao, 2008*). Recently, *Long et al., 2013b*, have experimentally identified regions with non-methylated DNA in seven diverse vertebrates. They called those regions non-methylated islands (NMIs). *Long et al., 2013b*, demonstrated that in some instances NMIs do not coincide with computationally classified CGIs (*Table 1*). Furthermore, they showed that NMIs, and not CGIs, are central to the definition of gene promoters in the vertebrates that they studied.

For understanding how CGIs and NMIs impact the local chromatin structure and contribute to gene regulation, it is important to know how DNA mechanics is influenced by its sequence and epigenetic modifications. (In this work, we are solely concerned with double-stranded or dsDNA, which we therefore just hereafter refer to as DNA.) One of the widely studied properties of DNA is the sequence-dependent effects on nucleosome positioning. A nucleosome comprises 147 base pairs of DNA wrapped around the histone core and is the elementary unit of DNA packing into chromatin. The positions and dynamics of nucleosomes contribute to DNA transcription, replication, and repair (*Andrews and Luger, 2011*; *Yasuda et al., 2005*; *Chen et al., 2010*). Various computational models have been developed for predicting nucleosome positioning based on DNA sequence (*Loshikhes et al., 2006*; *Segal et al., 2006*; *Gupta et al., 2008*; *Struhl and Segal, 2013*), physical properties (*Gabdank et al., 2009*; *Gabdank et al., 2010*), and deformation free energy (*Ruscio and Onufriev, 2006*; *Battistini et al., 2010*; *Chen et al., 2016*; *Eslami-Mossallam et al., 2016*; *Liu et al., 2018*). It has been shown that methylation and hydroxymethylation change DNA mechanical properties (*Pérez et al., 2012*; *Battistini et al., 2021*) and nucleosome-forming affinity (*Buitrago et al., 2021*; *Choy et al., 2010*; *Lee and Lee, 2012*; *Lee et al., 2015*; *Jimenez-Useche and Yuan, 2012*; *Li et al., 2022*). For example, *Ngo et al., 2016*, demonstrated that methylation of DNA decreases the mechanical stability of a nucleosome, as measured by a fluorescence-force spectroscopy assay. Whereas multiple studies reveal that DNA methylation induces a more compact and rigid nucleosome structure (*Choy et al., 2010*; *Lee and Lee, 2012*; *Lee et al., 2015*). Another computational study by *Yoo et al., 2021*, showed that DNA methylation of CpG sites can significantly increase the bending energy.

In this work, we compute the free energy, required for DNA to reach a configuration in a nucleosome, as well as the probability density, associated with the optimal nucleosomal configuration of DNA, for ensembles of sequence fragments drawn from different regions across the human genome, and compare with analogous computations on sequence ensembles generated artificially. To model sequence-dependent DNA mechanics, we use the *cgNA+* model (https://cgdnaweb.epfl.ch; *Sharma et al., 2023*; *De Bruin and Maddocks, 2018*).

In previous work, we presented a method for predicting a sequence-dependent configuration and associated free energy of DNA wrapped on a nucleosome (*Giniūnaitė and Petkevičiūtė-Gerlach, 2022*). The method is based on minimisation of the *cgNA+* model free energy for a given sequence while constraining the positions of phosphates bound to the histone core. The indices and allowed positions of bound phosphates were identified from the cylindrical coordinates of 30 experimental PDB structures of nucleosomes.

In this article, we use an improved version of this method to explore the differences in nucleosome wrapping energies and the probability densities for nucleosomal configurations between sequences drawn from inside and outside both CGIs and NMIs. We first show that the nucleosome wrapping energy increases with increasing concentration of CpG dinucleotide steps only when the cytosines in those steps are methylated or hydroxymethylated. Then, we investigate intersections and disjunctions of CGI and NMI regions and demonstrate that the intersection of these two sequence ensembles

ensures the lowest probability densities of nucleosomal configurations. We also show that the probability densities of nucleosomal configurations decrease with increasing CpG numbers. Finally, we investigate the relation between wrapping energies and experimentally observed nucleosome occupancy scores (*Schwartz et al., 2019*; *Yazdi et al., 2015*).

## Methods
### The cgNA+ model

*cgNA+* is a coarse-grained model of double-stranded nucleic acids (dsNA). A linear dsNA is modelled as a system of rigid bases and phosphates, and its configuration is described by a coordinate vector $w \in \mathbb{R}^N$. Given an arbitrary $n$ base pair sequence $\mathcal{S}$ and a model parameter set $\mathcal{P}$, *cgNA+* constructs the expected, or ground, or minimum energy configuration $\mu(\mathcal{S}, \mathcal{P}) \in \mathbb{R}^N$ and the (banded) stiffness, or inverse covariance, matrix $K(\mathcal{S}, \mathcal{P}) \in \mathbb{R}^{N \times N}$ with $N = 24\,n - 18$, scaled such that

$$U(\mathrm{w}; \mathcal{S}, \mathcal{P}) := \frac{1}{2} \left( \mathrm{w} - \mu \right) \cdot K \left( \mathrm{w} - \mu \right) \tag{1}$$

is the energy (or the free energy difference between the configurations $w$ and $\mu$) expressed in units of kT. Then,

$$\rho(\mathrm{w}; \mathcal{S}, \mathcal{P}) := \frac{1}{Z} \exp \left\{ -\mathrm{U}(\mathrm{w}; \mathcal{S}, \mathcal{P}) \right\} \tag{2}$$

is an equilibrium distribution on coordinates $w$ in the Gaussian, or multidimensional normal, form. Here, $Z$ is the normalising constant, or partition function,

$$Z = (2\pi)^{\frac{N}{2}} \det(K)^{-\frac{1}{2}}. \tag{3}$$

In this presentation, we restrict the parameter set $\mathcal{P}$ to cases describing DNA with arbitrary sequences in the alphabet {A, T, C, G, MpN, HpK}, where MpN and HpK are CpG dinucleotide steps in which the cytosines are either both methylated or both hydroxymethylated, respectively.

The *cgNA+* model is an extension in two directions of the precursor *cgDNA* model (*Gonzalez et al., 2013*; *Petkevičiūtė et al., 2014*) in which the configuration coordinate $w$ was restricted to rescaled versions of the standard intra and inter base-pair Curves+ (*Lavery et al., 2009*) coordinates which determine the relative rigid body displacements of all the bases in a DNA (and which respect the Tsukuba convention; *Olson et al., 2001*). For our purposes, the first critical extension of *cgDNA* was to *cgDNA+* (*Patelli, 2019*) in which the coordinate vector $w$ was extended to explicitly include the relative rigid body displacements between bases and adjacent phosphate groups, also assumed to be rigid, but only with a parameter set $\mathcal{P}$ allowing sequences $\mathcal{S}$ in the standard {A, T, C, G} alphabet. The second crucial extension from *cgDNA+* to *cgNA+* (*Sharma et al., 2023*; *Sharma, 2023*) was to estimate, and test, parameter sets for other dsNAs and with extended alphabets, including epigenetically modified bases. In this presentation, we consider only the case of DNA but with a parameter set that distinguishes between unmodified CpG dinucleotide steps, methylated CpG dinucleotide steps (symmetrically so that both cytosines are modified, denoted MpN), and hydroxymethylated CpG dinucleotide steps (again symmetrically and denoted HpK).

*cgNA+* parameter sets are estimated by fitting model predictions for first and second moments (or respectively $\mu(\mathcal{S}, \mathcal{P})$ and $K^{-1}(\mathcal{S}, \mathcal{P})$) for a training library of sequences $\mathcal{S}_i$ to statistics drawn directly from large-scale, fully atomistic molecular dynamics (or MD) simulations. The MD simulation protocol reflects both assumed physical solvent conditions, such as counter ion species and concentration, and the choice of atomistic MD simulation potentials. The parameter set $\mathcal{P}$ adopted here was based on simulations with 150 mM KCl ions and the AMBER software (*Pearlman et al., 1995*; *Case et al., 2005*) with the parmbsc1 force field (*Ivani et al., 2016*), explicit TIP3P water (*Jorgensen et al., 1983*), and the *Joung and Cheatham, 2008*, ion model. The additional MD force field parameters for modified cytosines were taken from *Pérez et al., 2012*, and *Battistini et al., 2021*. MD simulations of twelve 24 base-pair length sequences were used for training model parameters for methylated DNA. These sequences contained methylated CpG steps and combinations of methylated CpG steps in diverse sequence contexts. An analogous training library was used to train hydroxymethylated DNA

parameters. The model parameters for unmodified DNA were separately trained on a diverse and comprehensive library of 16 sequences containing all possible tetranucleotides at least once.

The predictions of the *cgNA+* model were found to be extremely accurate compared to an extensive set of test MD simulations and in good agreement with limited experimental protein-DNA X-ray crystallography data (*Sharma, 2023*). Above all, the *cgNA+* model is computationally so efficient that predictions of statistics for hundreds of thousands of sequences can be easily handled, which is not feasible with direct MD simulation. Thus, we used *cgNA+* free energy for linear fragments as the starting point for developing a method for computing sequence-dependent nucleosome wrapping energies.

## Nucleosome wrapping energy for a DNA sequence

A sequence-dependent configuration $w_{opt}$ of 147 bp of DNA wrapped into a nucleosome is modelled by minimising the *cgNA+* energy $U(w; \mathcal{S}, \mathcal{P})$ (*Equation 1*):

$$w_{opt}(\mathcal{S}, \mathcal{P}) = \arg\min_{W} \left( U(w; \mathcal{S}, \mathcal{P}) + \sum_{i=1}^{28} C_i(w) \right) \tag{4}$$

where

$$C_i(\mathbf{w}) = c_i \left\| p_i(\mathbf{w}) - \bar{p}_i \right\|^2, \quad i = 1, \ldots, 28 \tag{5}$$

is a set of elastic constraints on the positions $p_i(w)$ of the 28 DNA phosphates, which are closest to the histone core. $\mathcal{S}$ is any given DNA sequence of length 147 bp and $\mathcal{P}$ is our *cgNA+* model parameter set. The reference positions $\bar{p}_i$ were obtained from a set of 100 experimental PDB structures of nucleosomes by averaging, and the indices of the 28 phosphates, closest to the nucleosome core, are identified as in our previous work (*Giniūnaitė and Petkevičiūtė-Gerlach, 2022*). The penalty coefficients $c_i$ are set through numerical experiments to keep the distances $\|p_i(\mathbf{w}_{opt}) - \bar{p}_i\|$ within the ranges observed in the PDB structures, while avoiding steric clashes between the two turns of DNA in a nucleosome.

The energy minimisation (*Equation 4*) is performed numerically using the *fminunc* function of MATLAB with provided gradient and Hessian values. An averaged configuration of 100 experimental structures of DNA in nucleosomes is used as the initial or starting configuration for the optimisation procedure for all sequences. The optimisation for a 147 bp DNA sequence takes approximately 30 s.

The energy minimisation algorithm used in this work improves its previous version (*Giniūnaitė and Petkevičiūtė-Gerlach, 2022*), which was sensitive to the starting configuration and reached minimum energies with inflated magnitude while keeping the trends similar to experimental observations. This development improves those shortcomings by incorporating two main changes. Firstly, the constraints (*Equation 5*) are elastic, in contrast to previously used hard intervals. In addition, rather than performing the optimisation (*Equation 4*) in the *cgNA+* coordinates and computing the absolute positions of the constrained phosphates after every minimisation step, we use a mixed coordinate vector, with absolute positions of constrained phosphates, absolute positions and orientations of their adjacent bases, and all the base pairs, while the rest of the configuration is described in the *cgNA+* coordinates. Although the conversion to the full *cgNA+* coordinates for evaluating the energy is still necessary after every optimisation step, this approach provides the possibility to derive the gradient vector and the Hessian matrix for the constrained optimisation problem (*Equation 4*), which significantly improves the performance of the algorithm. As a consequence of these modifications, nucleosome wrapping energies are similar in magnitude, as well as in trends to those observed in experiments (as discussed in Results).

In this work, we compare sequence-dependent energy values $U(w_{opt}; \mathcal{S}, \mathcal{P})$ (*Equation 1*) with units kT, as well as the natural logarithms of the optimal nucleosomal configuration probability density (*Equation 2*)

$$\ln \rho(w_{opt}; \mathcal{S}, \mathcal{P}) = -U(w_{opt}; \mathcal{S}, \mathcal{P}) + \frac{1}{2} \ln \det(K)$$
$$- \frac{N}{2} \ln(2\pi) \tag{6}$$

for different sequences $\mathcal{S}$. The probability densities can be regarded as proportional to probabilities of DNA spontaneously reaching the configuration $\mathrm{w_{opt}}$, when these probabilities are estimated in a small domain around $\mathrm{w_{opt}}$, with the same domain volume for all the sequences. The negative log probability density, $-\ln \rho(\mathrm{w_{opt}}; \mathcal{S}, \mathcal{P})$, is equivalent to the free energy associated with the configuration $\mathrm{w_{opt}}$. It is also worth noting that

$$\ln \rho(\mathrm{w_{opt}}; \mathcal{S}, \mathrm{P}) = -U(\mathrm{w_{opt}}; \mathcal{S}, \mathcal{P}) - H(\mathcal{S}, \mathcal{P}) + \frac{N}{2}, \tag{7}$$

where $H(\mathcal{S}, \mathcal{P})$ is the entropy.

A more detailed mathematical description of the computational method will be published separately, and the MATLAB code is available at https://github.com/daivaaviad/optDNA_nucleosome (copy archived at *Petkevičiūtė-Gerlach, 2025*).

## Experimental data

Computationally predicted CGI regions from the human genome are obtained from the UCSC genome browser (*Kent et al., 2002*), whereas experimentally identified NMIs for human liver cells are taken from *Long et al., 2013b*. The human genome version used in these studies is Genome Reference Consortium Human Build 37 (GRCh37). Note that to make the necessary computations feasible, for each specific sequence in an ensemble such as CGIs, NMIs, or their intersections or complements, we only consider one specific central 147 bp sequence per region. The exact sequences used in our analysis are available at https://github.com/rginiunaite/CGI-NMI-sequences. Data for nucleosome occupancy scores for HeLa cells was taken from *Schwartz et al., 2019*, and for human genome embryonic stem cells from *Yazdi et al., 2015*.

## Results

### The spread of predicted DNA nucleosomal configurations is similar to that of experimental structures

We first compare our predicted sequence-dependent optimal DNA nucleosomal configurations (*Equation 4*) for 100 human genome sequences with 100 experimental configurations from the Protein Data Bank (*Berman et al., 2000*). The human genome sequences are a random subset of our sequence sample for the CGI and NMI intersection in Chromosome 1, but the following observations remain unchanged for sequence samples from different genomic regions.

In *Figure 1a*, we observe an orderly positioning of phosphates in the aligned experimental structures. Note that, because the structures are aligned, the phosphates adjacent to the nucleosome dyad fall into the same spatial cluster despite the variation in sequence length across the PDB structures. For each helical turn, we choose one phosphate cluster that is closest to the nucleosome centre (points coloured in red) and use the index of that cluster to define the constraints in *Equation 5*. *Figure 1c* shows the analogous scatter plot for the configurations $\mathrm{w_{opt}}(\mathcal{S}, \mathcal{P})$ (*Equation 4*) predicted for the first 100 non-methylated CGI sequences in our human genome sample. The positioning of phosphates in *Figure 1c* is rather similar to the one in *Figure 1a*, and the clusters of phosphates are of comparable sizes in both plots, even though there seems to be more variation in the experimental structures. This difference can be explained by the diversity of experimental settings, such as differences in ion concentration, the presence of histone modifications, additional ligands, and other experimental conditions that are not captured in our model. The variation in predicted structures could be increased by reducing the penalty coefficients in *Equation 5*. However, this would require additional constraints in *Equation 4* to avoid the self-overlap of DNA (steric clashes between the two DNA turns in the nucleosome), which is not present with the current setting. Another difference between the two plots is the unwrapping of approximately five base pairs at each end of the predicted configurations. While in our model there are no restrictions for this behaviour, in the experimental setting there could be other factors, such as histone tails, keeping the DNA ends closer to the nucleosome core. This issue could be solved by adding additional constraints at the ends of the 147 bp sequence. Such a modification would increase the nucleosome wrapping energy only marginally, as it would affect about 10 of the 147 base pairs. Two side views of the experimental and predicted nucleosome structures are displayed in *Figure 1—figure supplement 1*. The plots, analogous to those in parts (c) and (d) of

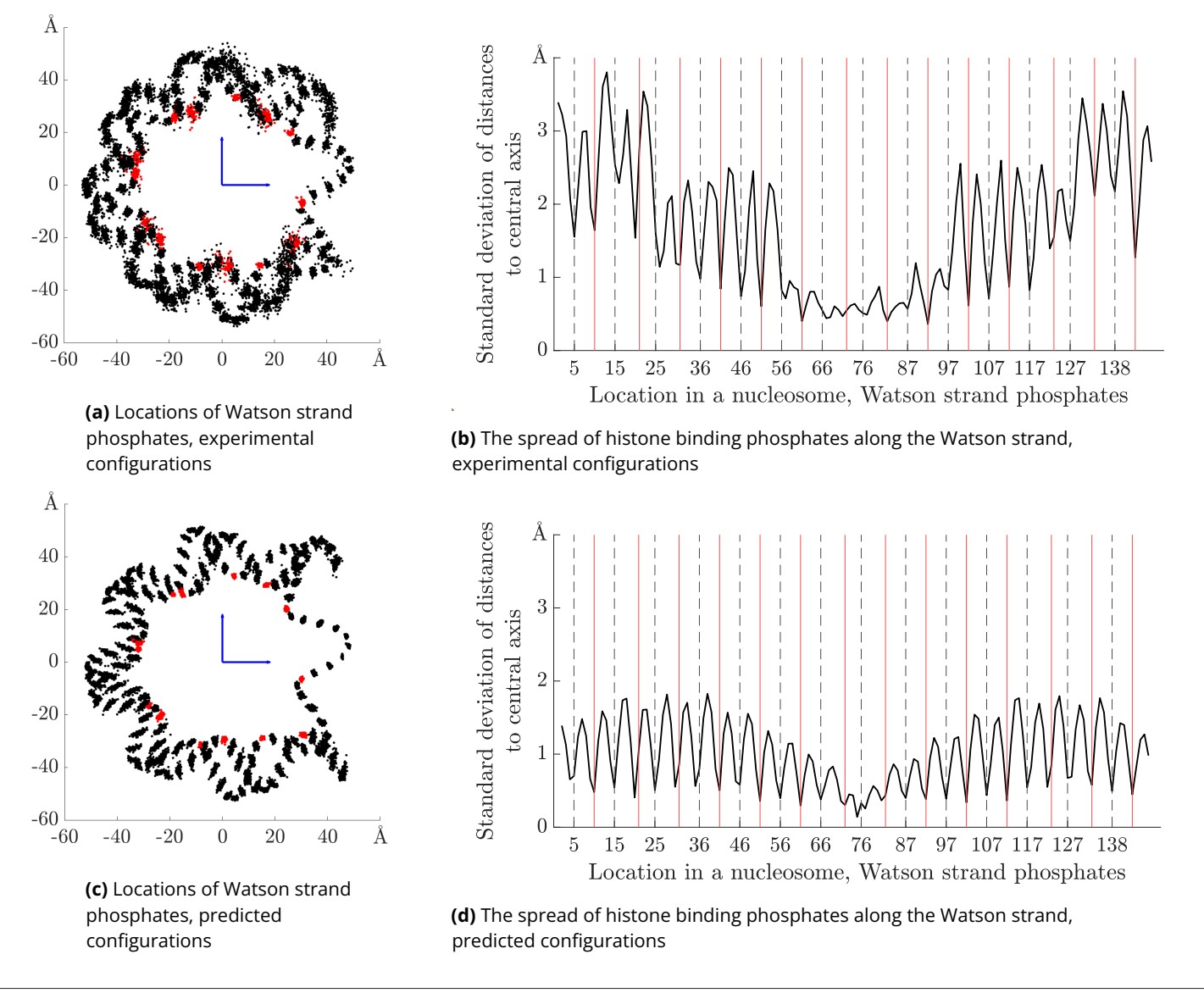

**(a)** Locations of Watson strand phosphates, experimental configurations

**(b)** The spread of histone binding phosphates along the Watson strand, experimental configurations

**(c)** Locations of Watson strand phosphates, predicted configurations

**(d)** The spread of histone binding phosphates along the Watson strand, predicted configurations

**Figure 1.** Locations and spread of phosphates in experimental versus predicted nucleosome structures. Left column: locations of the Watson strand phosphates for 100 aligned nucleosome structures, projected to a plane perpendicular to the nucleosome central axis. Top row corresponds to 100 experimental PDB nucleosome structures (not all with independent sequences). Red points are phosphates with local minima of radial distance used to identify bound indices. Bottom row analogous data over 100 predicted minimal energy nucleosomal configurations for sequences drawn from human genome CpG islands. The phosphates with bound indices that are constrained during the optimisation are coloured in red. Right panels: standard deviations over sequence of radial distance of all phosphates against index along the Watson strand. Top PDB structures, bottom model computations. Bound indices are marked with solid red vertical lines. Dashed black vertical lines mark indices of bound complementary (Crick) strand phosphates.

The online version of this article includes the following figure supplement(s) for figure 1:

**Figure supplement 1.** Locations of the Watson strand phosphates for 100 aligned nucleosome structures, projected to planes parallel to the nucleosome central axis (side views of the nucleosomes).

**Figure supplement 2.** Locations and spread of phosphates in nucleosome structures: a comparison of predictions for sequences from different human chromosomes.

*Figure 1*, but corresponding to sequences drawn from human chromosomes 2, 3, and 4, are displayed in *Figure 1—figure supplement 2*.

The spread of phosphate positions in each cluster is quantified by the standard deviations of phosphate distances to the nucleosome central axis, plotted in *Figure 1b and d*. As already seen in the scatter plots on the left, the spread of the predicted configurations is smaller and more regular than

that of the experimental structures. The main conclusion here is that the standard deviation reaches its local minima for the phosphate clusters closest to the nucleosome core (indices marked by solid red vertical lines, corresponding to the red points on the left plots). Interestingly, the local minima of standard deviations is also reached for positions corresponding to the histone touching phosphates on the complementary (Watson) strand (marked by dashed black vertical lines). This observation holds for both experimental and predicted nucleosomal configurations and indicates that the phosphates chosen to be constrained in our optimisation method are also constrained (bound to the histone core) in the experimental nucleosomes.

## CpG step (hydroxy)methylation affects DNA nucleosome wrapping energy and the probability density of nucleosomal configuration

To assess the sequence dependence of the nucleosome wrapping energy and of the probability density of the optimal nucleosomal configuration, we initially perform a computational experiment in which we generate four sets of sequences of length 147 bp, each containing a thousand sequences with a varying number of CpG dinucleotide steps, ranging from 0 to 4, from 5 to 14, from 15 to 24, and from 25 to 34. Each sequence is first generated with equal probabilities for each base, and then if the desired density of CpG steps needs to be increased, dinucleotide steps in random positions are replaced by CpGs. Similarly, if the density needs to be decreased, a base in a CpG dinucleotide is replaced by another, all in a randomised way. From these sequence ensembles, we also create another eight sets of sequences, first by symmetrically methylating (MpN), and second by hydroxymethylating (HpK), both cytosines in all the instances of the CpG dinucleotides.

We then use our optimisation algorithm to compute the energies required for these sequences to wrap onto nucleosomes. The resulting energy values are shown in *Figure 2*. The average of the predicted nucleosome wrapping energy over all the 4K unmodified random sequences is 86.12 kT. As expected, this value is higher than the energy prediction for the synthetic nucleosome positioning sequence Widom 601 (*Lowary and Widom, 1998*) (76.23 kT) and the naturally occurring sequence 5S, known to have a high nucleosome forming affinity (*Simpson and Stafford, 1983*) (83.76 kT). An opposite extreme, the 147 bp poly-A sequence, has a high predicted wrapping energy of 95.08 kT. Above examples illustrate that the modelling matches expectations for some known DNA sequences. When we vary unmodified CpG density, only minor differences in wrapping energy are observed (*Figure 2a*). However, the average energy increases substantially when cytosines are methylated or hydroxymethylated to obtain MpN or HpK steps. These results can be well associated with the findings that suggest that methylation increases DNA stiffness (*Lee and Lee, 2012*; *Pérez et al., 2012*; *Ngo et al., 2016*). The effects of hydroxymethylation and methylation are quite similar.

The changes in nucleosome wrapping energy due to CpG methylation or hydroxymethylation can be explained not only by altered DNA stiffness, but also by modifications in its equilibrium configuration. For example, the roll, twist, and slide inter base-pair coordinates are strongly affected when DNA wraps onto a nucleosome (*Giniūnaitė and Petkevičiūtė-Gerlach, 2022*) and they are all substantially modified in the linear ground state when cytosines are methylated or hydroxymethylated (*Figure 3a*). The linear ground state coordinates of the phosphates also change both when wrapping onto a nucleosome and with cytosine modification (*Figure 3b*), but this change is more dependent on the sequence context (*Sharma, 2023*). The same observation holds for intra base-pair coordinates (*Figure 3—figure supplement 1*). The ground state changes resulting from cytosine modifications – primarily characterised by an average increase in roll and a decrease in twist – may be linked to steric hindrance caused by the cytosine 5-substituent (*Battistini et al., 2021*). Notably, the negative coupling between twist and roll has already been observed in X-ray crystallography data (*Olson et al., 1998*).

We then compare the values of the logarithm of the probability density of the optimal nucleosomal configurations (*Equation 6*). The probability density is proportional to the probability of DNA spontaneously acquiring its optimal nucleosomal configuration, estimated in a small domain around that configuration. It can also be regarded as a measure of DNA mechanical affinity to form nucleosomes, which includes the (negative) nucleosome wrapping energy and also approximates entropic effects or thermal fluctuations.

For our set of random sequences, the log probability density decreases with the growing number of unmodified CpG steps (*Figure 2c*). Cytosine methylation weakens the trend while also increasing

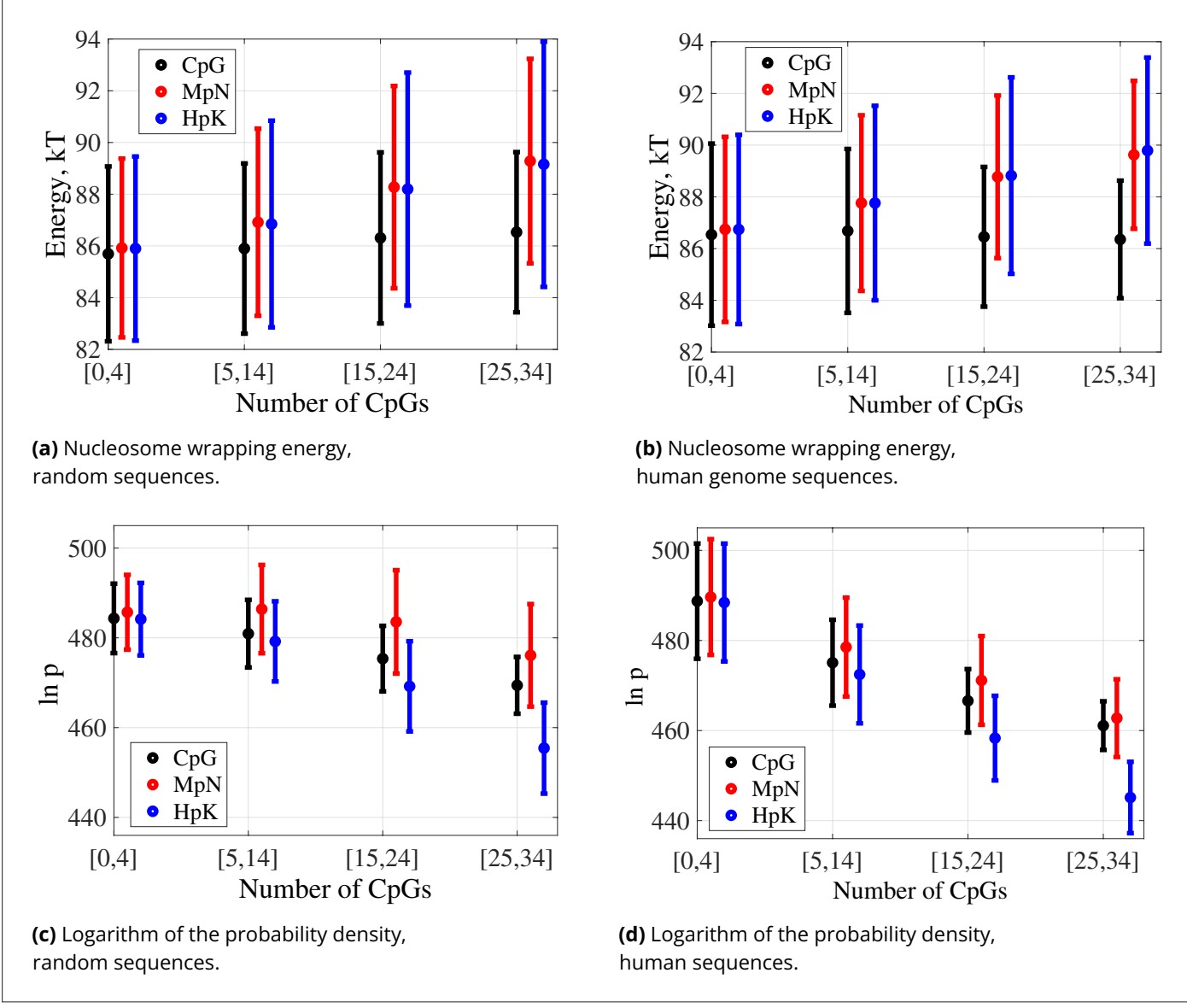

**(a)** Nucleosome wrapping energy, random sequences.

**(b)** Nucleosome wrapping energy, human genome sequences.

**(c)** Logarithm of the probability density, random sequences.

**(d)** Logarithm of the probability density, human sequences.

**Figure 2.** Spectra of nucleosome wrapping energies and logarithms of probability densities for the optimal nucleosomal configurations for 147 bp sequences (**a, c**) generated randomly and (**b, d**) drawn from the human genome, grouped by the indicated ranges of numbers of CpG dinucleotide steps: dots averages, bars standard deviation in sequence. For methylated and hydroxymethylated data, all CpG steps are symmetrically modified. Numbers of sequences falling into each CpG range are given in *Table 2*.

The online version of this article includes the following figure supplement(s) for figure 2:

**Figure supplement 1.** Spectra of nucleosome wrapping energies and logarithms of probability densities for the optimal nucleosomal configurations for 147 bp human genome sequences, grouped by the indicated ranges of numbers of CpG dinucleotide steps: dots averages, error bars standard deviation in sequence.

**Figure supplement 2.** Distances between CpG dinucleotides when there are 10 CpG dinucleotides in sequences of length 147.

**Figure supplement 3.** Spectra of (**a**) nucleosome wrapping energy and (**b**) natural logarithms of probability densities for DNA nucleosomal configurations for unmethylated (CpG), methylated (MpN), and hydroxymethylated (HpK) DNA sequences with CpG dinucleotide count from 5 to 14.

the average log densities within each range of CpG count. In contrast, cytosine hydroxymethylation leads to a faster decrease in log densities with the growing CpG count.

To verify whether our observations for randomly generated sequences also hold for biologically more realistic sequences, we perform the same analysis for sequences obtained from the human genome. We consider four sub-ensembles of our human sequence fragments grouped by their

**Table 2.** Numbers of human genome sequence fragments of length 147 bp taken from CpG islands (CGIs), non-CGIs, non-methylated islands (NMIs), and non-NMIs grouped by the number of CpG dinucleotide steps in each of four intervals.
As expected, CGI fragments have relatively more CpG junctions than non-CGI fragments. NMIs also have more CpGs than non-NMIs.

| CpG count | [0, 4] | [5, 14] | [15, 24] | [25, 34] |
|---|---|---|---|---|
| CGI | 216 | 16392 | 10248 | 814 |
| Not CGI | 43512 | 15272 | 340 | 76 |
| NMI | 7624 | 19383 | 9308 | 796 |
| Not NMI | 36104 | 12281 | 1280 | 94 |

numbers of CpG dinucleotides falling in the intervals that correspond to constrained numbers of CpG steps in our randomised sequence ensembles. *Figure 2b* demonstrates that for the human sequence ensembles, just as for the random sequence ensembles (*Figure 2a*), the nucleosome wrapping energy is not strongly affected with the number of unmodified CpG dinucleotide steps. Cytosine (hydroxy) methylation also increases the nucleosome wrapping energy for human genome sequences. However, some differences can be observed between the two ensembles. For most of the human sequence sub-ensembles, there are somewhat higher nucleosome wrapping energies and a sharper drop in log probability densities than for the comparable random ensembles. This observation remains unchanged after sub-sampling human genome sequences to have 1K data points in each CpG range, the same number as for random sequences (*Figure 2—figure supplement 1*). We first hypothesised that different clustering features of CpG dinucleotides might explain these differences. To investigate this hypothesis, we looked at the distances along the sequences between CpG dinucleotides. But we did not observe any significant differences in the distributions of these distances between human genome and random sequence ensembles (*Figure 2—figure supplement 2*).

These differences might instead be associated with non-random distribution of other dinucleotide steps in the two sets of sequence ensembles. *Figure 4* gives the average number of all of the 16

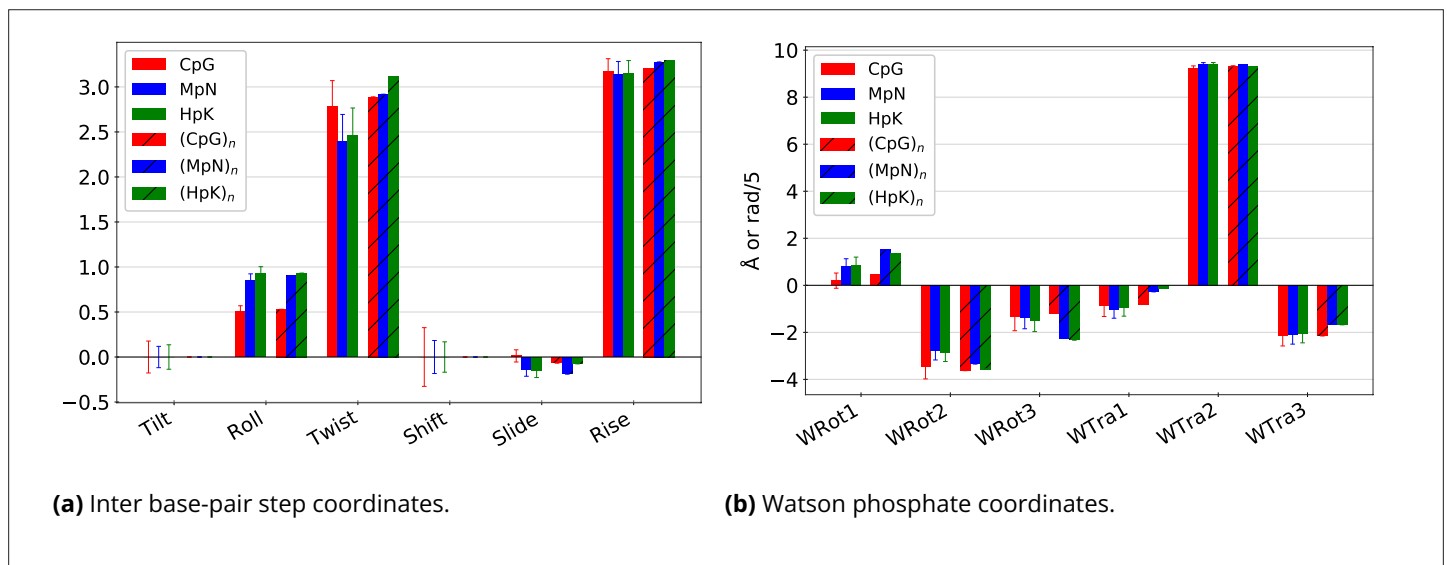

**(a)** Inter base-pair step coordinates.

**(b)** Watson phosphate coordinates.

**Figure 3.** Effects of sequence context and epigenetic base modifications on the cgNA+ model predicted ground state shape of CpG steps. Statistics over $4^8$=65,536 sequences of 22 bp length, constructed around the central CpG step as GCGTCGX4X3X2X1CGY1Y2Y3Y4GTCGGC, with all the possible $Xj$ and $Yj \in \{A, T, C, G\}$, $\forall j \in \{1, 2, 3, 4\}$. Bar plots show the ground state values of (**a**) six inter base-pair step and (**b**) six Watson phosphate coordinates for CpG steps (i) averaged over sequence context with standard deviations in thin lines and (ii) the extreme case of poly(CpG) (in hatch). In each case, three versions corresponding to unmodified, methylated, and hydroxymethylated steps. The standard deviations highlight the crucial role of non-local sequence dependence in the equilibrium structure of CpG/MpN/HpK steps. Analogous plots for the remaining intra base-pair coordinates and Crick phosphate coordinates are shown in *Figure 3—figure supplement 1*.

The online version of this article includes the following figure supplement(s) for figure 3:

**Figure supplement 1.** Effects of sequence context and epigenetic base modifications on the ground state shape of CpG steps.

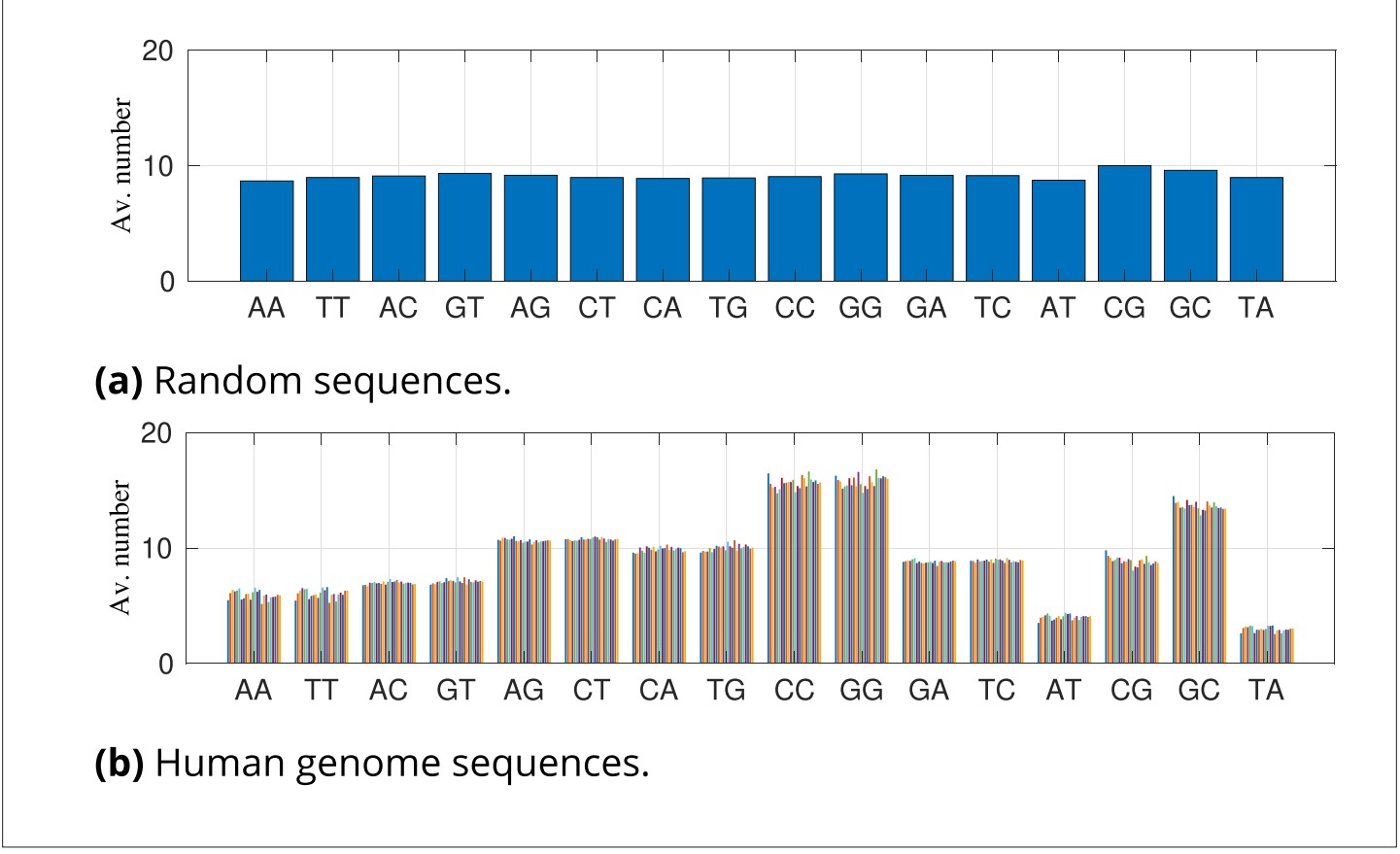

**(a)** Random sequences.

**(b)** Human genome sequences.

**Figure 4.** Average number of instances of the 16 different dinucleotide steps for (**a**) 1000 random 147 bp sequences and for (**b**) our 147 bp human genome sequence ensemble, with [5, 14] CpGs. Different colours in (**b**) correspond to fragments taken from different chromosomes. Dinucleotide steps are ordered next to their complements, with self-complementary steps listed on the right.

The online version of this article includes the following figure supplement(s) for figure 4:

**Figure supplement 1.** Average number of the 16 different dinucleotide steps for (**a**) 1000 random sequences and for (**b**) our human genome sequence ensemble, with [0, 4] CpGs.

**Figure supplement 2.** Average number of the 16 different dinucleotide steps for (**a**) 1000 random sequences and for (**b**) our human genome sequence ensemble, with [15, 24] CpGs.

**Figure supplement 3.** Average number of the 16 different dinucleotide steps for (**a**) 1000 random sequences and for (**b**) our human genome sequence ensemble, with [25, 34] CpGs.

possible dinucleotide steps in the random and human ensembles in the case of 10 CpGs for random and [5, 14] CpGs for human ensembles (other cases are provided in *Figure 4—figure supplements 1–3*). The distribution can be seen to be highly non-uniform for the human genome sequences. For example, one striking feature of the [5, 14] human sequence ensemble is the small number of ApT and TpA dinucleotides. In fact, ApT and TpA are found to be the most stiff and flexible dimer steps in both experiments and simulations (*Young et al., 2022*; *Sharma, 2023*). It may well be that this depletion is a result of promoter sequences avoiding mechanically extreme dimer steps.

We further tested whether the non-uniform dinucleotide counts, as opposed to the specific arrangement of dinucleotides, are the key reason for the difference in energies and nucleosomal configuration probability densities between the human and random sequences. To this end, we explored the scenario in which we keep the same count of each dinucleotide step in each sequence in the [5, 14] human genome sequence ensemble, but we reordered the dinucleotide steps using the Altschul-Erickson dinucleotide shuffle algorithm (*Altschul and Erickson, 1985*). We observe that in this scenario, the resulting distributions of nucleosome wrapping energies and of nucleosomal configuration log probability densities remain significantly more similar to that of the unshuffled human ensemble than to the analogous random sequence ensemble (*Figure 2—figure supplement*

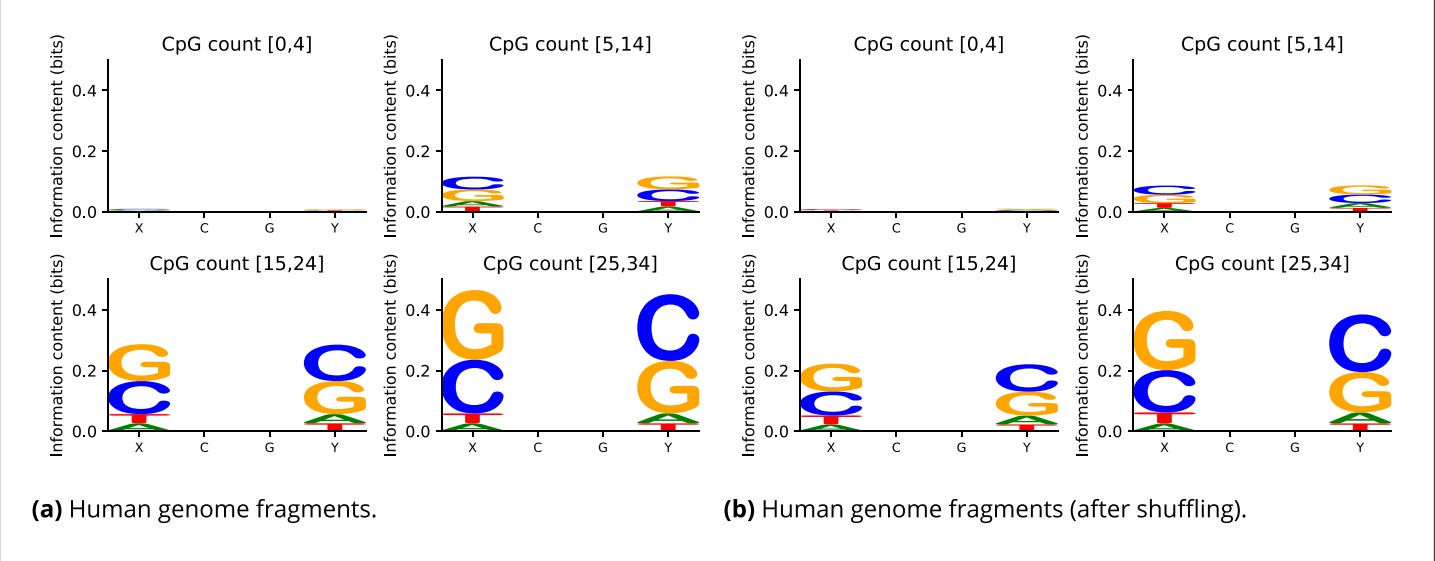

**(a)** Human genome fragments.

**(b)** Human genome fragments (after shuffling).

**Figure 5.** Sequence logos for tetramer flanking context of CpG dinucleotide steps for (**a**) all four sequence ensembles from the human genome with varying numbers of CpG junctions, and (**b**) all four sequence ensembles from the human genome after dinucleotide shuffling (but respecting the numbers of dinucleotide steps). Just specifying the numbers of CpG dinucleotide steps is a strong enough constraint to leave the tetramer sequence context logos largely unchanged after shuffling. The sequence logos in panel (**a**) for the human sequence ensemble before sequence shuffling suggest a slightly stronger C/G flanking enrichment than after shuffling.

*3*). This observation suggests that the non-uniform count of dinucleotides is central in explaining the differences in wrapping energies and log probability densities between random and human genome sequence ensembles.

In fact, the ground state configuration of the DNA in each junction has a quite strong dependence on sequence context beyond the junction dinucleotide. This phenomenon has been observed in MD simulation (*Pasi et al., 2014*; *Balaceanu et al., 2019*) and crystallography experiments (*Young et al., 2022*). It is also encapsulated in the *cgNA+* model. It has further been observed (*Sharma, 2023*) that epigenetic base modifications lead to larger changes in the ground state configuration within CpG junctions when the two flanking bases in the tetramer context are C/G rich (also *Figure 3*). For instance, in an average context, hydroxy(methylation) of CpG steps reduces its twist significantly. In contrast, when a poly-CpG sequence is hydroxy(methylated), the predicted twist of the CpG steps increases (*Figure 3*). Therefore, for assessing the effect of sequence shuffling on the ground shape of DNA, it is of interest to investigate the flanking context of the CpG dinucleotides. The tetranucleotide sequence logos over all CpG steps in three of our four sub-ensembles of human sequences are in fact rich in C/Gs, as shown in the sequence logos of *Figure 5a*, where the amount of the flanking enrichment depends on the four cases of ranges of numbers of CpG dinucleotide steps. It is also the case that the constraints on the elevated number of CpG steps in the fragments are strong enough that the tetranucleotide sequence logos remain essentially unaltered in each of the four cases for the sequence ensembles that arise after the dinucleotide step sequence shuffling algorithm is applied (*Figure 5b*). Nevertheless, when comparing the logos in panels (a) and (b) in detail, there is a signal indicating that the flanking C/G enrichment is slightly stronger in the original human sequence ensemble than it is after shuffling.

## Overlap of CGIs and NMIs leads to the lowest probability densities of nucleosomal configurations

In this section, we split the human genome into four regions based on data from *Long et al., 2013b*: (A) intersection of CGIs and NMIs; (B) NMIs that do not intersect with CGIs; (C) CGIs that do not intersect with NMIs; (D) regions that intersect neither with CGIs nor with NMIs. The numbers of sequences in each sub-ensemble are listed in *Table 1*.

The data presented in *Figure 6a* reveals that the nucleosome wrapping energies have similar distributions in all four regions, if we do not include methylation (round dots and solid error bars). If we

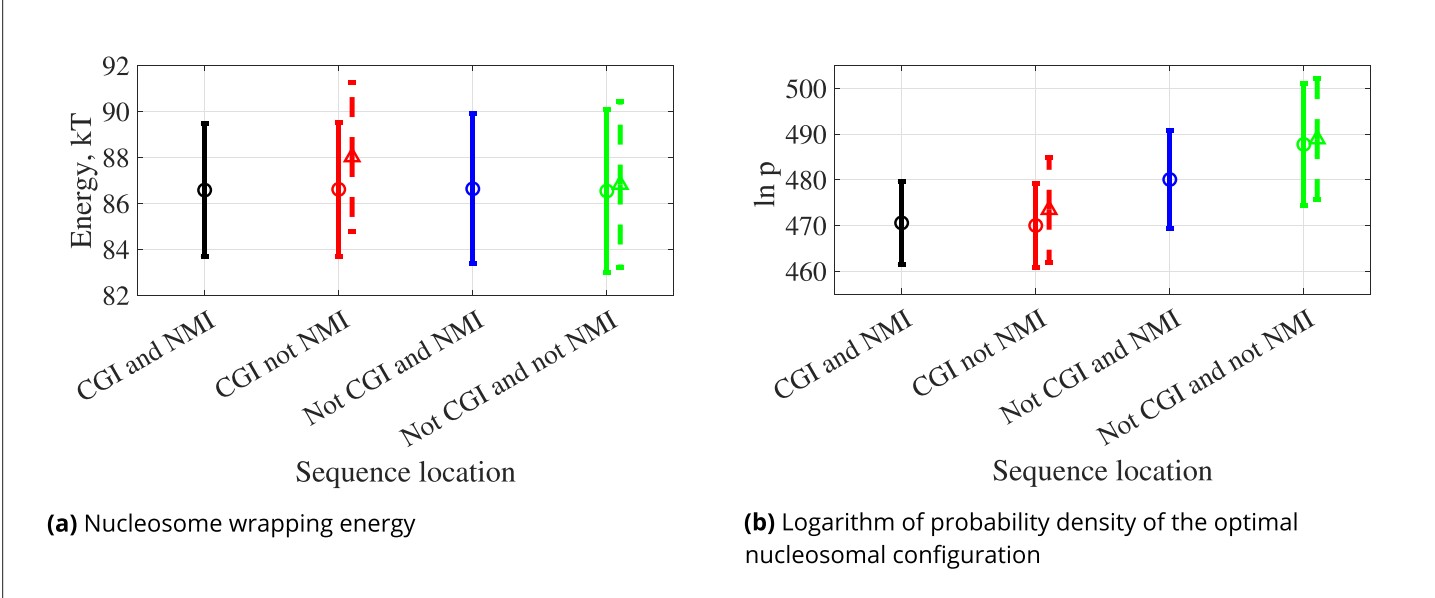

(a) Nucleosome wrapping energy

(b) Logarithm of probability density of the optimal nucleosomal configuration

**Figure 6.** Spectra of (**a**) nucleosome wrapping energies and (**b**) log probability densities of the optimal nucleosomal configurations for 147 bp sequences drawn from four different regions of the human genome: (A) intersection of CpG island (CGI) and non-methylated island (NMI), (B) NMI and not CGI, (C) CGI and not NMI, (D) not CGI and not NMI (*Table 1*). Dots represent averages, error bars represent standard deviation over sequence, solid and circles when CpG dinucleotides are not methylated, dashed and triangles when CpGs are methylated.

include methylation everywhere in not NMIs (i.e. respecting the definition of NMI), there is an increase in the wrapping energy for sequences that are CGIs that are not NMIs (triangle dots and dashed error bars in *Figure 6a*, red). Wrapping energies for sequences that belong neither to CGIs nor to NMIs do not exhibit such a significant change upon methylation (green).

The log probability density of the optimal nucleosomal configuration has the lowest average value for sequences in the intersection of CGIs and NMIs. Even though methylation of the sequences that are CGIs but not NMIs increases the log probability density values, the highest densities are for sequences that are not CGIs but are NMIs (blue) or in the regions outside CGIs and NMIs (green).

It is important to note that the number of sequences drawn from the four different regions is not equidistributed. *Table 1* shows that there are fewer sequences that are CGIs but not NMIs, i.e., they are methylated CGIs, than in the other three categories. Nevertheless, approximately 30% of CGIs are methylated, so it is reasonable to consider methylated CGIs as a separate category. Note that for practical restrictions on total computational resources, we compute wrapping energies for only one 147 bp representative sequence drawn from each occurrence of each of the four types of regions over the entire genome. *Table 1* reports the resulting numbers of fragments, i.e., the number of instances of each of the four types of regions. But the numbers in *Table 1* do not reflect the total number of bp covered by each of the four types of region. In reality, the number of base pairs in each occurrence of the regions that are neither CGI nor NMI is much higher than in the other three types, so that the union of all not CGI and not NMI regions covers by far most of the genome.

## Nucleosome wrapping energies and probability densities of the optimal nucleosomal configurations compared with nucleosome occupancy scores

We now compare our wrapping energy predictions and DNA nucleosomal configuration log probability density predictions with experimentally measured nucleosome occupancy scores as reported in *Schwartz et al., 2019* (an experiment with HeLa cells) human genome and *Yazdi et al., 2015* (an experiment with embryonic stem cells). We have extracted their reported occupancy scores for each of our selected 147 bp fragments and first grouped the data by the methylated and non-methylated regions (NMIs and not NMIs), then within each region according to the number of CpGs in the corresponding sequences.

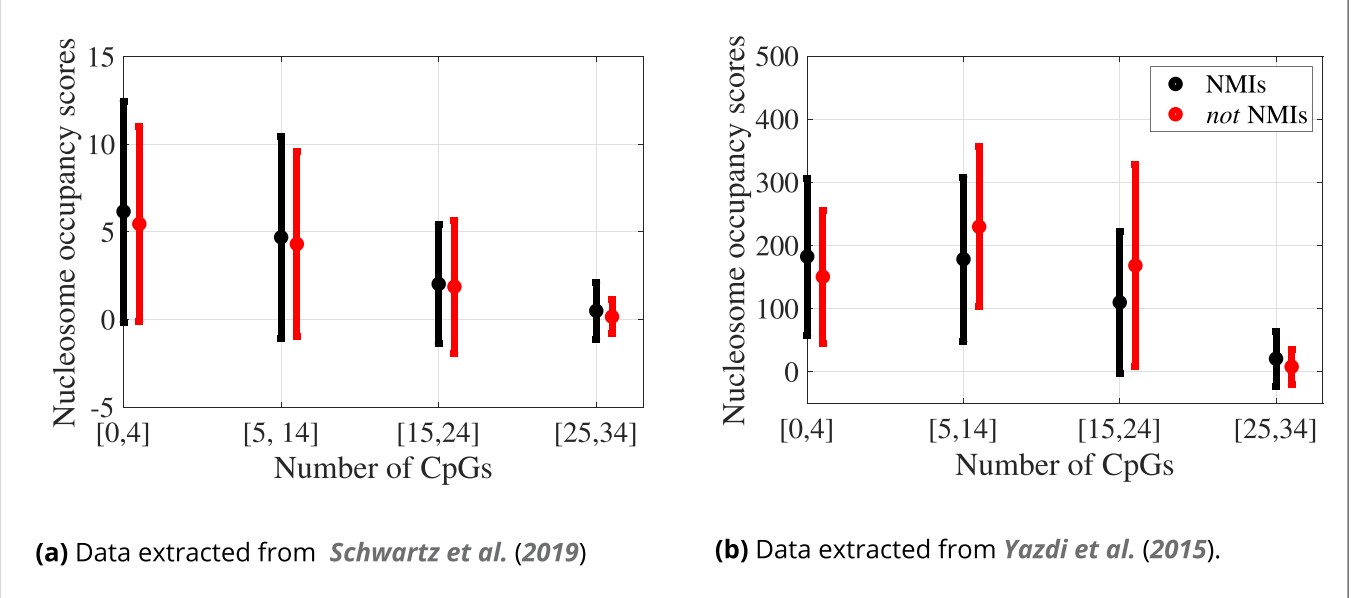

**(a)** Data extracted from *Schwartz et al.* (*2019*)

**(b)** Data extracted from *Yazdi et al.* (*2015*).

**Figure 7.** Spectra of nucleosome occupancy scores for our 86,874 selected sequences, grouped by the genomic regions (non-methylated island [NMI] and *not* NMIs) and by indicated ranges of numbers of CpG dinucleotide steps: dots averages, error bars standard deviation in sequence. The number of sequences in each group is listed in *Table 2*. See also *Figure 2d*.

*Figure 7* shows that nucleosome occupancy is decreasing with increasing CpG count for both NMI and not NMI regions, with one exception of passing from [0, 4] to [5, 14] in the Yazdi et al. data. This trend is compatible with the increase in nucleosome wrapping energy for methylated sequences in *Figure 2b* and the decrease in log probability density for nucleosomal configurations in *Figure 2d*.

According to Yazdi et al. data, for the CpG count falling into the middle intervals, from 5 to 14 and from 15 to 24, methylated sequences have a higher average occupancy than unmethylated sequences. This difference is also observed in our log probability density predictions in *Figure 2d*. For

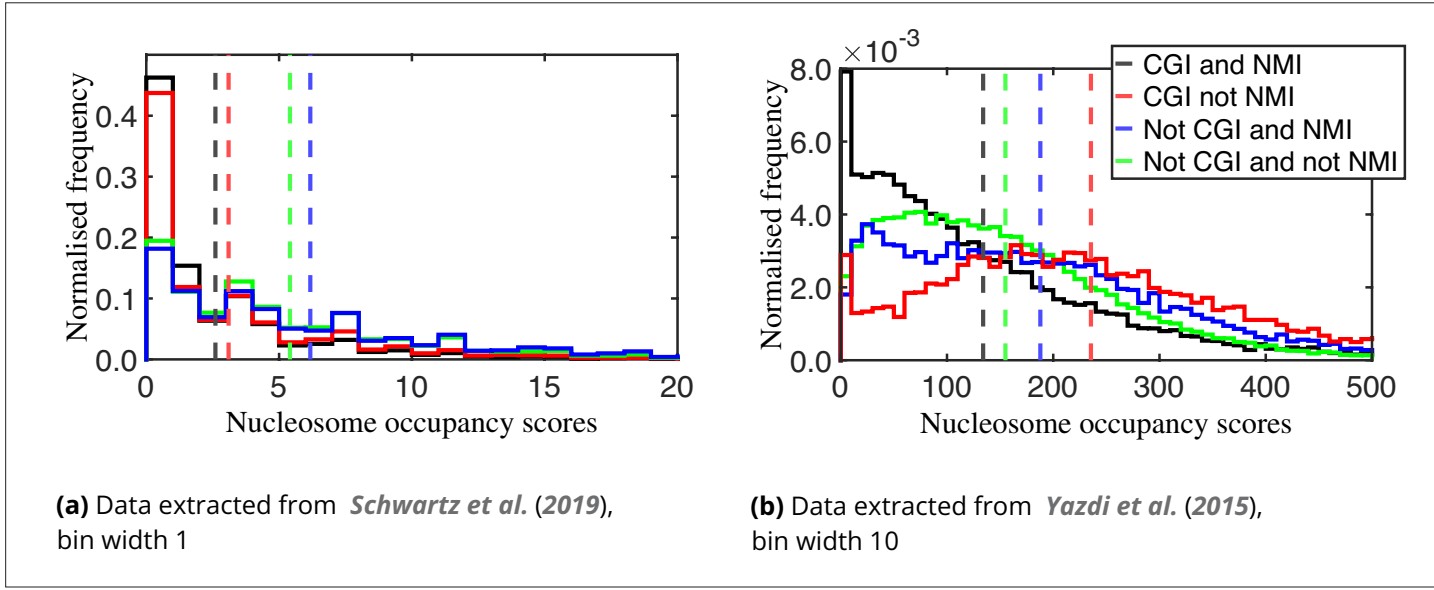

**(a)** Data extracted from *Schwartz et al.* (*2019*), bin width 1

**(b)** Data extracted from *Yazdi et al.* (*2015*), bin width 10

**Figure 8.** Normalised frequencies (each of the four histograms in each plot normalised independently) of experimental nucleosome occupancy scores for our 86,874 selected sequences grouped by each of the four types of regions in the genome (*Table 1*). Average score for each region is indicated by a vertical dashed line of appropriate colour. The black and red (but not blue or green) histograms have significant spikes reflecting many instances of zero occupancy in the experimental data.

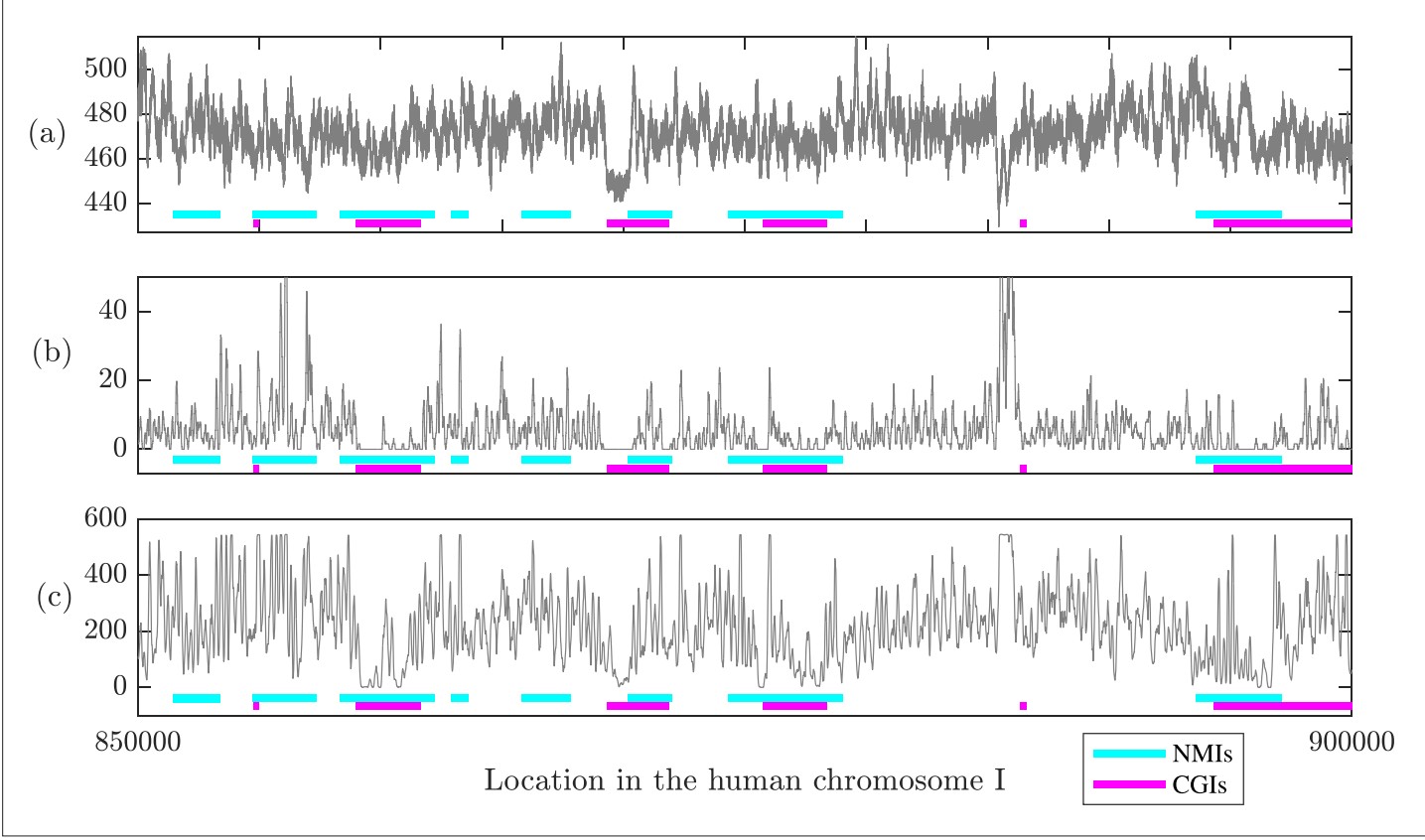

**Figure 9.** Predicted log probability density for an optimal nucleosomal configuration (**a**) , nucleosome occupancy scores from *Schwartz et al., 2019* (**b**) , and nucleosome occupancy scores from *Yazdi et al., 2015* (**c**) , for sequence positions 850K–900K of human chromosome I. In the regions corresponding to the intersection of CpG islands (CGIs) and non-methylated islands (NMIs), both the mean log probability density (468.61) and mean scores (2.62 and 139.53) are smaller than outside of the intersection regions (476.10, 5.89, and 212.00, respectively).

the remaining two CpG count intervals and all the Schwartz et al. data, the occupancy for methylated sequences is lower than or very similar to unmethylated ones.

We then grouped our sequences according to the four genomic sub-regions. *Figure 8a* reveals that for both sets of data, the average of nucleosome occupancy scores is lowest for the intersection of CGIs and NMIs (black). For the data extracted from *Schwartz et al., 2019*, methylated CGIs (red) have a higher average nucleosome occupancy than unmethylated CGIs, but smaller than the non-CGI regions. For *Yazdi et al., 2015*, data, the distribution of nucleosome occupancy scores is highest for the intersection of CGIs and not NMIs (red), i.e., both the lowest and highest occupancy distributions arise for sequences drawn from CGIs, with the lowest occupancies in unmethylated fragments and the highest in methylated fragments. All these observations are statistically significant, as demonstrated in *Supplementary file 1A-D*.

Both sets of experimental data indicate that in CGIs the highest occupancies arise for the fragments that have methylated CpG dinucleotides and therefore higher nucleosome wrapping energies. This conclusion, in particular, apparently runs counter to the (perhaps naive) intuition that high nucleosome-forming affinity should arise for fragments with low wrapping energy. Instead, a higher log probability density seems to be a better indicator of higher occupancy scores: the lowest average of log probability densities corresponds to the unmethylated CGIs (*Figure 6b*).

In order to further probe this observation, we selected a 50K run of bp in the human genome. (Specifically from chromosome I, between genomic positions 850K and 900K, as this range contains the largest number of CGI and NMI intersections.) We then computed the probability density of an optimal nucleosomal configuration for every possible 147 bp window at the resolution of 1 bp shifts. (These computations are quite intensive, requiring around 900 hr of CPU time for the relatively short 50K bp segment, which is why longer subsequences were not considered.) The resulting data is

plotted in *Figure 9a*, with the CGIs indicated with magenta underlining and NMIs in cyan. On average, the lowest log probability densities arise at the intersections of CGIs and NMIs: the mean value of log probability density is 468.61 kT over the intersection of CGI and NMI regions, and 476.10 kT in the complementary regions.

Panels (b) and (c) of *Figure 9* provide analogous plots for occupancy scores, again taken from *Schwartz et al., 2019*, and *Yazdi et al., 2015*, respectively. Again, the lowest average values arise for sequences in the intersection of CGIs and NMIs: the average scores are 2.62 and 139.53 in the intersection of CGIs and NMIs, versus 5.89 and 212.53 outside of the intersection regions.

The observations about nucleosome occupancy should be regarded as preliminary and be treated with caution, as they are based on experimental data obtained for the cancerous HeLa cells (*Schwartz et al., 2019*) and human genome embryonic stem cells (*Yazdi et al., 2015*), while for the classification of NMI and not NMI, we use the data of *Long et al., 2013b*, obtained from human liver cells. Nevertheless, since the lowest log probability densities in the human genome are predicted for CpG-rich sequences regardless of their methylation state (*Figure 2d*), and the same holds for both sets of the nucleosome occupancy scores (*Figure 7*), we conclude that the lowest occupancies occur for sequences with the lowest log probability densities.

## Conclusions

In this work, we studied the computed sequence-dependent mechanical nucleosome wrapping energy, required to deform a linear 147 bp DNA fragment to a configuration, where the appropriate 28 phosphates can bind to the histone core, as well as the probability density function, that can be regarded as proportional to the probability of linear DNA spontaneously reaching the nucleosomal configuration.

We explored the sequence dependence of the energy and the probability density corresponding to our predicted optimal nucleosomal DNA configurations. Our analysis includes the effects of both methylation and hydroxymethylation epigenetic modifications of CpG dinucleotides. To achieve this, we used the newly developed computational method to solve the constrained minimisation problem (*Equation 4*) in terms of the *cgNA+* energy (*Equation 1*) subject to constraints on the phosphates binding to histones in given ranges of configurations. The fact that the *cgNA+* model includes an explicit description of the phosphate group configurations allows for a comparatively simple description of the DNA-histone binding site constraints, which we believe to be a significant improvement over prior rigid base-pair coarse-grained DNA models used for nucleosome wrapping energy prediction (*Eslami-Mossallam et al., 2016*; *Chen et al., 2016*; *Liu et al., 2018*; *Neipel et al., 2020*). We believe that our minimisation algorithm delivers an accurate ordering of sequence-dependent wrapping energies and probability densities, given the accuracy of the *cgNA+* energy (*Equation 1*). The *cgNA+* probability density function (*Equation 2*) is itself known to deliver highly accurate sequence-dependent statistics of linear fragments compared to MD simulations carried out with the same protocol as the *cgNA+* parameter set training data. However, an MD protocol perfectly emulating experimental conditions (which are often different in different experiments) is challenging and therefore, some approximations must be made. For example, the parameter set used here models DNA in 150 mM KCl solution, whereas both ion type and concentration might be different in both experiment and in vivo.

Nucleosome wrapping energies, the corresponding optimal configurations, and their probability densities could also be computed via approaches that adopt MD simulations directly, e.g., (*Ruscio and Onufriev, 2006*; *Ngo et al., 2016*; *Battistini et al., 2021*). Along with accurate treatment of sequence-dependent mechanics of DNA, the key advantage of our coarse-grained approach is that it is computationally much more efficient, so that large numbers of sequences can be considered. For example, when epigenetic sequence variants are included, the data described in this article involves approximately 400K solves of the minimisation problem (*Equation 4*). And analogous numbers of MD simulations are currently unfeasible.

The minimisation principle (*Equation 4*) delivers not only a wrapping energy and a probability density, but also the detailed configuration $w_{opt}$ realising the minimal wrapping energy. We compared our computed optimal configurations of DNA in a nucleosome with the experimental PDB structures and found significant similarities between the two configuration ensembles. Further and more detailed analysis is both feasible and interesting. For example, the roll and slide (inter base-pair coordinates)

are strongly affected when DNA wraps onto a nucleosome (*Giniūnaitė and Petkevičiūtė-Gerlach, 2022*), and they are both substantially modified in the linear ground state when cytosines are methylated or hydroxymethylated (*Sharma, 2023*). The linear ground state coordinates of the phosphates also change with cytosine modification, but this change is more dependent on the sequence context (*Sharma, 2023*).

We then computed spectra of wrapping energies and the nucleosomal configuration probability densities for ensembles of 147 bp fragments with differing numbers of CpG dinucleotides, with sequences both generated artificially and drawn from the human genome. We concluded that for increasing numbers of CpG steps, the wrapping energies increased substantially, but only for epigenetically modified CpGs. The effects on the wrapping energies of the two epigenetic modifications of methylation and hydroxymethylation are very similar. The nucleosomal configuration probability densities decreased with increasing CpG counts for both unmodified and (hydroxy)methylated DNA. However, for each CpG count interval, methylation increased and hydroxymethylation decreased the average probability densities.

As discussed fully in the main text, these trends were similar in both the artificial and human genome sequence ensembles, although there are perceptible differences, perhaps because of local and non-local sequence dependence in DNA. Notably, the two data sets have different flanking contexts, e.g., the human genome sequences have a small bias towards having more C/G flanking bases in the tetramer context to central CpG dinucleotides, along with some highly non-uniform distributions of other dinucleotides, e.g., very low occurrences of ApT and TpA steps.

We then compared nucleosome wrapping energies, in both epigenetically unmodified and modified versions, for ensembles of DNA sequences constructed by drawing one representative from each instance in the human genome of the four region types CGI and NMI, CGI and not NMI, not CGI and NMI, and finally not CGI and not NMI. We were motivated to consider four types of region by the work of *Long et al., 2013b*, who demonstrated that NMIs cannot be reliably identified by CGIs algorithms and NMIs may have more biological significance. They also found that NMIs are consistent across species, and in warm-blooded organisms these regions coincide with transcription initiation sites. The assumption that CGIs never have epigenetically modified CpG dinucleotides is often made when analysing CGIs (*Ioshikhes and Zhang, 2000*; *Hannenhalli and Levy, 2001*; *Bock et al., 2007*; *Han and Zhao, 2008*), although the current definitions of CGIs do not actually entail this information, so that the studies often lack detail in this respect (*Long et al., 2013b*). Accordingly, we considered all four possibilities of intersections and disjunctions between CGIs and NMIs. Our main conclusion from studying wrapping energy spectra from the four types of region is that the lowest probability densities of nucleosomal configurations arise precisely for unmodified CGI sequences, i.e., sequences that are both CGI and NMI.

The restriction to drawing one representative from each instance of each of the four types of region was dictated merely to limit the necessary computations to a feasible magnitude. We did verify that our results were not sensitive to precisely how we chose the 147 bp representative from each region. Another limitation dictated by available computational resources is the focus on human genome data only. It would be interesting to explore the same data (CGIs and NMIs) for other warm- and cold-blooded organisms which were also provided by *Long et al., 2013b*. That data might provide deeper insights because the regions of interest and their intersections differ vastly across different organisms.

## Discussion

We believe that our predictive computational model of nucleosome wrapping energies and the nucleosomal configuration probability densities is (subject to the aforementioned caveats) both sufficiently accurate and efficient to explore biologically pertinent ensembles of sequences and compare model predictions with experimental observations. It is presumably the case that nucleosome wrapping energy will make a significant contribution to predicting nucleosome binding affinities at a particular site. Both stiffness and ground state of a DNA fragment (which are accurately captured in the *cgNA+* model *Sharma, 2023*; *Sharma et al., 2023*) contribute to the sequence dependence of wrapping energy. At the same time, differences in stiffness also contribute to sequence-dependent differences in fluctuations about the minimal energy wrapped configuration $w_{opt}$. Thus, we believe that sequence (including epigenetic modifications) dependent entropy-like corrections are necessary to be able to

accurately predict binding affinities from wrapping energies, and computing the probability densities of the optimal nucleosomal configurations is a way to account for those corrections.

Furthermore, the process of comparing the predicted densities with the nucleosome occupancy scores is fraught with many potential sources of inaccuracy. Firstly, any computation involving only the DNA takes no account of the possibly sequence-dependent contributions of the histone tails, epigenetically modified or not. Secondly, the probability densities are not probabilities of DNA wrapping into nucleosomal configurations, but could be regarded as proportional to such, assuming that these probabilities can be approximated by a one-point integral over a small domain of the same volume for all the sequences. The validity of this assumption is not completely obvious.

Generally, there have been opposing views in the literature about the relationship between nucleosome occupancy scores and sequence-induced mechanical properties of DNA. *Pérez et al., 2012*, showed that genomic regions with high wrapping energy are nucleosome-depleted. *Yoo et al., 2021*, claimed that nucleosome occupancy scores anticorrelate with the wrapping energy. In contrast, it has been shown that CGIs are fivefold depleted for observed nucleosome coverage (*Valouev et al., 2011*), suggesting a positive correlation between nucleosome binding energy and nucleosome occupancy scores. The effect of DNA methylation on nucleosome formation also remains debated. *Pérez et al., 2012*, and *Battistini et al., 2010*, found that methylation increases DNA deformation energy and decreases nucleosome formation. Similarly, (*Ngo et al., 2016*) showed that methylation decreases nucleosome stability. On the other hand, *Collings and Anderson, 2017*, demonstrated that methylated regions are among the highest nucleosome occupied elements in the genome. The conflicting results may reflect differences in experimental conditions and the contribution of cellular factors other than DNA mechanics to nucleosome formation in vivo. For example, *Pérez et al., 2012*, *Battistini et al., 2021*, and *Ngo et al., 2016*, derived their conclusions from experiments using modified Widom 601 sequences, while *Collings and Anderson, 2017*, is a whole-genome methylation study.

In this work, we contribute to this discussion by investigating the relations between our probability density predictions and the experimentally observed human genome nucleosome occupancy scores from *Schwartz et al., 2019*, and *Yazdi et al., 2015*. Our predictions agree with both sets of data in concluding that methylation of CGIs increases the probability of nucleosome formation. However, the precise ordering of the four genomic regions of CGI and NMI groups by nucleosome occupancy is different in all three cases (two experimental data sets and our predictions). This might be due to different methylation patterns for cancerous HeLa cells in *Schwartz et al., 2019*, human embryonic stem cells in *Yazdi et al., 2015*, and liver cells in *Long et al., 2013b*, used for identifying non-methylated regions for our computations. Matched DNA modification to nucleosome occupancy experimental data and investigation of different cell types will likely reveal more accurately how cells evolve nucleotide composition and modification patterns to reach optimal nucleosome occupancy in different genomic regions.

## Acknowledgements

This work received financial support from the Research Council of Lithuania (LMTLT), agreement number S-MIP-21-5 (for DP-G and RG), Marius Jakulis Jason fund (for RG), Swiss National Science Foundation, Grant Number 200020_182184 (for JHM and RS, as well as the EPFL Scitas computational resources used for this work), and Ludwig Cancer Research, Oxford (for SK).

## Additional information

### Funding

| Funder | Grant reference number | Author |
| --- | --- | --- |
| Lietuvos Mokslų Akademija | agreement number S-MIP-21-5 | Rasa Giniūnaitė Daiva Petkevičiūtė-Gerlach |
| Marius Jakulis Jason Fund | | Rasa Giniūnaitė |
| Swiss National Science Foundation | Grant Number 200020_182184 | Rahul Sharma John H Maddocks |

| Funder | Grant reference number | Author |
|---|---|---|
| Ludwig Cancer Research, Oxford | | Skirmantas Kriaucionis |

The funders had no role in study design, data collection and interpretation, or the decision to submit the work for publication.

## Author contributions
Rasa Giniūnaitė, Conceptualization, Data curation, Software, Formal analysis, Funding acquisition, Validation, Investigation, Visualization, Methodology, Writing – original draft; Rahul Sharma, Formal analysis, Investigation, Writing – original draft; John H Maddocks, Resources, Funding acquisition, Investigation, Methodology, Writing – original draft; Skirmantas Kriaucionis, Conceptualization, Investigation, Methodology, Writing – original draft; Daiva Petkevičiūtė-Gerlach, Conceptualization, Resources, Data curation, Software, Formal analysis, Supervision, Funding acquisition, Validation, Investigation, Visualization, Methodology, Writing – original draft, Project administration, Writing – review and editing

## Author ORCIDs
Rasa Giniūnaitė https://orcid.org/0000-0002-0538-6390
Rahul Sharma https://orcid.org/0000-0003-0700-3098
Skirmantas Kriaucionis https://orcid.org/0000-0002-2273-5994
Daiva Petkevičiūtė-Gerlach https://orcid.org/0000-0002-7646-2442

Reviewer #1 (Public review): https://doi.org/10.7554/eLife.98468.4.sa1
Reviewer #2 (Public review): https://doi.org/10.7554/eLife.98468.4.sa2
Reviewer #3 (Public review): https://doi.org/10.7554/eLife.98468.4.sa3
Author response https://doi.org/10.7554/eLife.98468.4.sa4

# Additional files

## Supplementary files
MDAR checklist

Supplementary file 1. Nucleosome wrapping energy in CpG islands and the role of epigenetic base modifications.

## Data availability
CGI regions from the human genome can be obtained from the UCSC genome browser (*Kent et al., 2002*), whereas experimentally identified NMIs for liver cells are provided in the SI of *Long et al., 2013b*. The human genome version used in these studies is Genome Reference Consortium Human Build 37 (GRCh37). The full listing of sequences used in our analysis are available in the github repository, at https://github.com/rginiunaite/CGI-NMI-sequences. The raw experimental nucleosome occupancy data can be accessed from the SI of *Schwartz et al., 2019* and *Yazdi et al., 2015*. The tables with the mean occupancy scores, difference of means and p-values corresponding to Figures 7 and 8 can be found in the *Supplementary file 1*.

The following dataset was generated:

| Author(s) | Year | Dataset title | Dataset URL | Database and Identifier |
|---|---|---|---|---|
| | | | | , |
| Giniūnaitė R | 2022 | CGI-NMI-sequences | https://github.com/rginiunaite/CGI-NMI-sequences | GitHub, 3053c66 |

The following previously published datasets were used:

| Author(s) | Year | Dataset title | Dataset URL | Database and Identifier |
|---|---|---|---|---|
| Schwartz U, Németh A, Diermeier S, Exler JH, Hansch S, Maldonado R, Heizinger L, Merkl R, Längst G | 2018 | Characterizing the nuclease accessibility of DNA in human cells to map higher order structures of chromatin | https://www.ncbi.nlm.nih.gov/geo/query/acc.cgi?acc=GSE100401 | NCBI Gene Expression Omnibus, GSE100401 |
| Yazdi PG, Pedersen BA, Taylor JF, Khattab OS, Chen YH, Chen Y, Jacobsen SE, Wang PH | 2015 | Nucleosome profiling in human embryonic stem cells | https://www.ncbi.nlm.nih.gov/geo/query/acc.cgi?acc=GSE49140 | NCBI Gene Expression Omnibus, GSE49140 |
| Long HK, Sims D, Heger A, Blackledge NP, Kutter C, Wright ML, Grützner F, Odom DT, Patient R, Ponting CP, Klose RJ | 2013 | Epigenetic conservation at gene regulatory elements revealed by non-methylated DNA profiling in seven vertebrates | https://www.ncbi.nlm.nih.gov/geo/query/acc.cgi?acc=GSE43512 | NCBI Gene Expression Omnibus, GSE43512 |

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
