## [Editor Report · eLife Assessment]

This **valuable** simulation study proposes a new coarse-grained model to explain the effects of CpG methylation on nucleosome wrapping energy. The model accurately reproduces the all-atom molecular dynamics simulation data, and the evidence to support the claims in the paper is **solid**. This work will be of interest to researchers working on gene regulation, mechanisms of DNA methylation and effects of DNA methylation on nucleosome positioning.

---

## [Referee Report · Reviewer #1 (Public review)]

In this manuscript, the authors used a coarse-grained DNA model (cgNA+) to explore how DNA sequences and CpG methylation/hydroxymethylation influence nucleosome wrapping energy and the probability density of optimal nucleosomal configuration. Their findings indicate that both methylated and hydroxymethylated cytosines lead to increased nucleosome wrapping energy. Additionally, the study demonstrates that methylation of CpG islands increases the probability of nucleosome formation.

The major strength of this method is that the model explicitly includes the phosphate group as DNA-histone binding site constraints, enhancing CG model accuracy and computational efficiency and allowing comprehensive calculations of DNA mechanical properties and deformation energies.

The revised version has addressed the concerns raised previously, significantly strengthening the study.

---

## [Referee Report · Reviewer #2 (Public review)]

Summary:

This study uses a coarse-grained model for double stranded DNA, cgNA+, to assess nucleosome sequence affinity. cgNA+ coarse-grains DNA on the level of bases and accounts also explicitely for the positions of the backbone phosphates. It has been proven to reproduce all-atom MD data very accurately. It is also ideally suited to be incorporated into a nucleosome model because it is known that DNA is bound to the protein core of the nucleosome via the phosphates.

It is still unclear whether this harmonic model parametrized for unbound DNA is accurate enough to describe DNA inside the nucleosome. Previous models by other authors, using more coarse-grained models of DNA, have been rather successful in predicting base pair sequence dependent nucleosome behavior. This is at least the case as long as DNA shape is concerned whereas assessing the role of DNA bendability (something this paper focuses on) has been consistingly challenging in all nucleosome models to my knowledge.

It is thus of major interest whether this more sophisticated model is also more successful in handling this issue. As far as I can tell the work is technically sound and properly accounts for not only the energy required in wrapping DNA but also entropic effects, namely the change in entropy that DNA experiences when going from the free state to the bound state. The authors make an approximation here which seems to me to be a reasonable first step.

Of interest is also that the authors have the parameters at hand to study the effect of methylation of CpG-steps. This is especially interesting as this allows to study a scenario where changes in the physical properties of base pair steps via methylation might influence nucleosome positioning and stability in a cell-type specific way.

Overall, this is an important contribution to the questions of how sequence affects nucleosome positioning and affinity. The findings suggest that cgNA+ has something new to offer. But the problem is complex, also on the experimental side, so many questions remain open. Despite of this, I highly recommend publication of this manuscript.

Strengths:

The authors use their state-of-the-art coarse grained DNA model which seems ideally suited to be applied to nucleosomes as it accounts explicitly for the backbone phosphates.

Weaknesses:

The authors introduce penalty coefficients c_i to avoid steric clashes between the two DNA turns in the nucleosome. This requires c_i-values that are so high that standard deviations in the fluctuations of the simulation are smaller than in the experiments.

---

## [Referee Report · Reviewer #3 (Public review)]

Summary:

In this study, authors utilize biophysical modeling to investigate differences in free energies and nucleosomal configuration probability density of CpG islands and nonmethylated regions in the genome. Toward this goal, they develop and apply the cgNA+ coarse-grained model, an extension of their prior molecular modeling framework.

Strengths:

The study utilizes biophysical modeling to gain mechanistic insight into nucleosomal occupancy differences in CpG and nonmethylated regions in the genome.

Weaknesses:

Although the overall study is interesting, the manuscripts need more clarity in places. Moreover, the rationale and conclusion for some of the analyses are not well described.

Comments on revised version:

The authors have addressed my concerns.

---

## [Author Response]

The following is the authors’ response to the original reviews.

**Reviewer #1 (Public Review):**
Summary:In this manuscript, the authors used a coarse-grained DNA model (cgNA+) to explore how DNA sequences and CpG methylation/hydroxymethylation influence nucleosome wrapping energy and the probability density of optimal nucleosomal configuration. Their findings indicate that both methylated and hydroxymethylated cytosines lead to increased nucleosome wrapping energy. Additionally, the study demonstrates that methylation of CpG islands increases the probability of nucleosome formation.Strengths:The major strength of this method is the model explicitly includes phosphate group as DNA-histone binding site constraints, enhancing CG model accuracy and computational efficiency and allowing comprehensive calculations of DNA mechanical properties and deformation energies.Weaknesses:A significant limitation of this study is that the parameter sets for the methylated and hydroxymethylated CpG steps in the cgNA+ model are derived from all-atom molecular dynamics (MD) simulations that use previously established force field parameters for modified cytosines (P´erez A, et al. Biophys J. 2012; Battistini, et al. PLOS Comput Biol. 2021). These parameters suggest that both methylated and hydroxymethylated cytosines increase DNA stiffness and nucleosome wrapping energy, which could predispose the coarse-grained model to replicate these findings. Notably, conflicting results from other all-atom MD simulations, such as those by Ngo T in Nat. Commun. 2016, shows that hydroxymethylated cytosines increase DNA flexibility, contrary to methylated cytosines. If the cgNA+ model were trained on these later parameters or other all-atom MD force fields, different conclusions might be obtained regarding the effects of methylated and hydroxymethylation on nucleosome formation.Despite the training parameters of the cgNA+ model, the results presented in the manuscript indicate that methylated cytosines increase both DNA stiffness and nucleosome wrapping energy. However, when comparing nucleosome occupancy scores with predicted nucleosome wrapping energies and optimal configurations, the authors find that methylated CGIs exhibit higher nucleosome occupancies than unmethylated ones, which seems to contradict the expected relationship where increased stiffness should reduce nucleosome formation affinity. In the manuscript, the authors also admit that these conclusions “apparently runs counter to the (perhaps naive) intuition that high nucleosome forming affinity should arise for fragments with low wrapping energy”. Previous all-atom MD simulations (P´erez A, et al. Biophys J. 2012; Battistini, et al. PLOS Comput Biol. 202; Ngo T, et al. Nat. Commun. 20161) show that the stiffer DNA upon CpG methylation reduces the affinity of DNA to assemble into nucleosomes or destabilizes nucleosomes. Given these findings, the authors need to address and reconcile these seemingly contradictory results, as the influence of epigenetic modifications on DNA mechanical properties and nucleosome formation are critical aspects of their study.Understanding the influence of sequence-dependent and epigenetic modifications of DNA on mechanical properties and nucleosome formation is crucial for comprehending various cellular processes. The authors’ study, focusing on these aspects, definitely will garner interest from the DNA methylation research community.

Training the *cgNA+* model on alternative MD simulation datasets is certainly of interest to us. However, due to the significant computational cost, this remains a goal for future work. The relationship between nucleosome occupancy scores and nucleosome wrapping energy is still debated, as noted in our Discussion section. The conflicting results may reflect differences in experimental conditions and the contribution of cellular factors other than DNA mechanics to nucleosome formation in vivo. For instance, Pérez et al. (2012), Battistini et al. (2021), and Ngo et al. (2016) concluded that DNA methylation reduces nucleosome formation based on experiments with modified Widom 601 sequences. In contrast, the genome-wide methylation study by Collings and Anderson (2017) found the opposite effect. In our work, we also use whole-genome nucleosome occupancy data.

Comments on revised version:The authors have addressed most of my comments and concerns regarding this manuscript.
**Reviewer #2 (Public Review):**
Summary:This study uses a coarse-grained model for double stranded DNA, cgNA+, to assess nucleosome sequence affinity. cgNA+ coarse-grains DNA on the level of bases and accounts also explicitly for the positions of the backbone phosphates. It has been proven to reproduce all-atom MD data very accurately. It is also ideally suited to be incorporated into a nucleosome model because it is known that DNA is bound to the protein core of the nucleosome via the phosphates.It is still unclear whether this harmonic model parametrized for unbound DNA is accurate enough to describe DNA inside the nucleosome. Previous models by other authors, using more coarse-grained models of DNA, have been rather successful in predicting base pair sequence dependent nucleosome behavior. This is at least the case as long as DNA shape is concerned whereas assessing the role of DNA bendability (something this paper focuses on) has been consistently challenging in all nucleosome models to my knowledge.It is thus of major interest whether this more sophisticated model is also more successful in handling this issue. As far as I can tell the work is technically sound and properly accounts for not only the energy required in wrapping DNA but also entropic effects, namely the change in entropy that DNA experiences when going from the free state to the bound state. The authors make an approximation here which seems to me to be a reasonable first step.Of interest is also that the authors have the parameters at hand to study the effect of methylation of CpG-steps. This is especially interesting as this allows to study a scenario where changes in the physical properties of base pair steps via methylation might influence nucleosome positioning and stability in a cell-type specific way.Overall, this is an important contribution to the questions of how sequence affects nucleosome positioning and affinity. The findings suggest that cgNA+ has something new to offer. But the problem is complex, also on the experimental side, so many questions remain open. Despite of this, I highly recommend publication of this manuscript.Strengths:The authors use their state-of-the-art coarse grained DNA model which seems ideally suited to be applied to nucleosomes as it accounts explicitly for the backbone phosphates.Weaknesses:The authors introduce penalty coefficients *c_i_* to avoid steric clashes between the two DNA turns in the nucleosome. This requires *c_i_*-values that are so high that standard deviations in the fluctuations of the simulation are smaller than in the experiments.

Indeed, smaller *c_i_* values lead to steric clashes between the two turns of DNA. A possible improvement of our optimisation method and a direction of future work would be adding a penalty which prevents steric clashes to the objective function. Then the *c_i_* values could be reduced to have bigger fluctuations that are even closer to the experimental structures.

**Reviewer #3 (Public Review):**
Summary:In this study, authors utilize biophysical modeling to investigate differences in free energies and nucleosomal configuration probability density of CpG islands and nonmethylated regions in the genome. Toward this goal, they develop and apply the cgNA+ coarse-grained model, an extension of their prior molecular modeling framework.Strengths:The study utilizes biophysical modeling to gain mechanistic insight into nucleosomal occupancy differences in CpG and nonmethylated regions in the genome.Weaknesses:Although the overall study is interesting, the manuscripts need more clarity in places. Moreover, the rationale and conclusion for some of the analyses are not well described.

We have revised the manuscript in accordance with the reviewer’s latest suggestions.

Comments on revised version:Authors have attempted to address previously raised concerns.
**Reviewer #1 (Recommendations for the authors):**
The authors have addressed most of my comments and concerns regarding this manuscript. Among them, the most significant pertains to fitting the coarse-grained model using a different all-atom force field to verify the conclusions. The authors acknowledged this point but noted the computational cost involved and proposed it as a direction for future work. Overall, I recommend the revised version for publication.
**Reviewer #2 (Recommendations for the authors):**
My previous comments were addressed satisfactorily.
**Reviewer #3 (Recommendations for the authors):**
Authors have attempted to address previously raised concerns. However, some concerns listed below remain that need to be addressed.(1) The first reviewer makes a valid point regarding the reconciliation of conflicting observations related to nucleosome-forming affinity and wrapping energy. Unfortunately, the authors don’t seem to address this and state that this will be the goal for the future study.

Training the *cgNA+* model on alternative MD simulation datasets remains future work. However, we revised the Discussion section to more clearly address the conflicting experimental findings in the literature on how DNA methylation influences nucleosome formation.

(2) Please report the effect size and statistical significance value for Figures 7 and 8, as this information is currently not provided, despite the authors’ claim that these observations are statistically significant.

This information is now presented in Supplementary Tables S1-S4.

(3) In response to the discrepancy in cell lines for correlating nucleosome occupancy and methylation analyses, the authors claim that there is no publicly available nucleosome occupancy and methylation data for a human cell type within the human genome. This claim is confusing, as the GM12878 cell line has been extensively characterized with MNaseseq and WGBS.

We thank the reviewer for this remark. We have removed the statement regarding the lack of data from the manuscript; we intend to examine the suggested cell line in future research.

(4) In response to my question, the authors claimed that they selected regions from chromosome 1 exclusively; however, the observation remains unchanged when considering sequence samples from different genomic regions. They should provide examples from different chromosomes as part of the supplementary information to further support this.

The examples of corresponding plots for other nucleosomes are now shown in Supplementary Figure S9.